# Selective enhancement of optical nonlinearity in two-dimensional organic-inorganic lead iodide perovskites

F.O. Saouma[1], C.C. Stoumpos [2], J. Wong[1], M.G. Kanatzidis[2] & J.I. Jang [1,3]

Reducing the dimensionality of three-dimensional hybrid metal halide perovskites can improve their optoelectronic properties. Here, we show that the third-order optical non-linearity, $n_2$, of hybrid lead iodide perovskites is enhanced in the two-dimensional Ruddlesden-Popper series, $(CH_3(CH_2)_3NH_3)_2(CH_3NH_3)_{n-1}Pb_nI_{3n+1}$ ($n = 1–4$), where the layer number ($n$) is engineered for bandgap tuning from $E_g = 1.60$ eV ($n = \infty$; bulk) to 2.40 eV ($n = 1$). Despite the unfavorable relation, $n_2 \propto E_g^{-4}$, strong quantum confinement causes these two-dimensional perovskites to exhibit four times stronger third harmonic generation at mid-infrared when compared with the three-dimensional counterpart, $(CH_3NH_3)PbI_3$. Surprisingly, however, the impact of dimensional reduction on two-photon absorption, which is the Kramers-Kronig conjugate of $n_2$, is rather insignificant as demonstrated by broadband two-photon spectroscopy. The concomitant increase of bandgap and optical nonlinearity is truly remarkable in these novel perovskites, where the former increases the laser-induced damage threshold for high-power nonlinear optical applications.

[1] Department of Physics, Applied Physics and Astronomy, State University of New York (SUNY) at Binghamton, P.O. Box 6000, Binghamton, NY 13902, USA. [2] Department of Chemistry, Northwestern University, 2145 Sheridan Road, Evanston, IL 60208, USA. [3] Department of Physics, Sogang University, Seoul 04107, South Korea. Correspondence and requests for materials should be addressed to J.I.J. (email: jjoon@binghamton.edu)

Three-dimensional (3D) organic–inorganic hybrid perovskites have emerged from the perspective of efficient photovoltaics owing to their excellent optoelectronic properties[1–3]. These materials also demonstrated great potential for light emission[4, 5] and nonlinear optical (NLO) applications[6–10]. Recently, it was shown that reducing the dimensionality of the system can further improve the longevity of perovskite solar cells[11–13] and light-emission performance[14] as well. Although strong third-order optical nonlinearity was reported in conventional two-dimensional (2D) perovskites[15], the effect is associated with an exciton resonance. In fact, similar effects were observed from some 3D lead halide perovskites, where third-order NLO effects are resonantly enhanced at the exciton or subgap state[9, 10]. However, a typical perovskite far away from such resonance generally exhibits nominal nonlinearity[6], which can be well understood by an existing two-band model[16, 17]. Clearly, the dimensional effect on perovskite nonlinearity has not been systematically investigated yet.

The third-order NLO susceptibility, $\chi^{(3)}$, is a complex quantity that every material possesses in which its real and imaginary parts are inherently related by causality, a Kramers-Kronig relation[16, 17]. The former is related to the nonlinear refractive index, $n_2 \propto \mathrm{Re}[\chi^{(3)}/n_0]$, and the latter to the two-photon absorption (2PA) coefficient, $\beta \propto \omega\mathrm{Im}[\chi^{(3)}/n_0]$, respectively, where $n_0$ is the linear refractive index of a material and $\omega$ is the optical frequency of light under strong light-matter interaction[17]. The development of new $\chi^{(3)}$ materials is essential for the advancement of nonlinear optics, especially for applications that require high power and longer wavelengths throughout the infrared (IR)[18, 19]. Either $n_2$ or $\beta$ can be utilized for specific purposes, but their coexistence in a single $\chi^{(3)}$ material often causes undesirable effects that limit the performance. For instance, a large $n_2$ effect is required for self-focusing applications but an inherently related large $\beta$ value lowers the laser-induced damage threshold (LIDT), which ultimately leads to optical damage of the material via efficient 2PA[20]. This fundamental limit arises because of the exclusive interdependence of $n_2$ nonlinearity and LIDT on the bandgap of the material. A similar issue also severely restricts the discovery of efficient second-order NLO materials for wave mixing[21]. Namely, a narrower (wider) bandgap results in a stronger (weaker) nonlinearity but a lower (higher) LIDT. This is especially serious for organic-based NLO materials whose applications are greatly limited by low LIDTs[22]. Some researchers describe this tradeoff as a balance[23, 24], and as such, simultaneously targeting a large nonlinearity and a high LIDT remains an outstanding problem. Nevertheless, next-generation NLO materials must overcome inter-dependence of these two critical NLO parameters in order to bring effective capabilities aiming at high-power/high-efficiency NLO applications.

Here, we demonstrate that $n_2$ nonlinearity and LIDT can be improved concurrently by tailoring the benchmark hybrid lead iodide perovskite, $CH_3NH_3PbI_3$ ($MAPbI_3$), along the (110) direction into the 2D direct-gap Ruddlesden-Popper series, $(BA)_2(MA)_{n-1}Pb_nI_{3n+1}$ [$BA = CH_3(CH_2)_3NH_3$ and $n = 1, 2, 3, 4$][11–13]. Upon reducing the perovskite layer number ($n$) of the 3D perovskite, $MAPbI_3$ ($n = \infty$), these homologous 2D perovskites undergo progressive bandgap blueshift, from $E_g = 1.60$ eV ($n = \infty$) to 1.89 eV ($n = 4$), 2.00 eV ($n = 3$), 2.14 eV ($n = 2$), and 2.40 eV ($n = 1$) due to dimensional reduction of the perovskite spacer layers[12]. Such a significant bandgap increase is beneficial for enhancing the LIDT via less efficient 2PA ($\beta \propto E_g^{-3}$), but may hamper the $n_2$ value according to the relation, $n_2 \propto E_g^{-4}$[17]. Despite the adverse $E_g^{-4}$ dependence, these wide-bandgap perovskites possess unusually large $n_2$ nonlinearity, which significantly outperforms $MAPbI_3$ and even a standard NLO chalcogenide, $AgGaSe_2$[25], both having narrower bandgaps. In

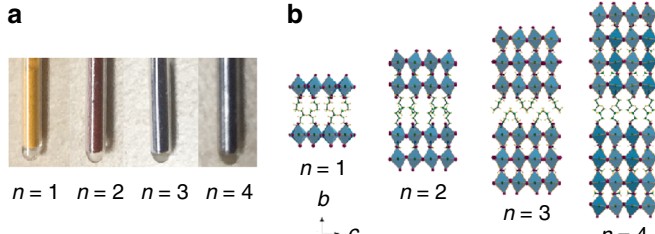

**Fig. 1** Two-dimensional perovskites and the corresponding crystal structures. **a** Photograph of sample powders in capillary tubes, showing bandgap blueshift upon decreasing the layer number, $n$. **b** Crystal structures viewed along the (100) crystallographic projection. The elements in the crystals are represented by different colors: Pb in *dark green*, I in *purple*, N in *dark blue*, H in *yellow*, and C in *light green*

fact, they show high performance when compared with typical semiconductors with $E_g \geq 1.0$ eV. This striking effect arises most likely from dimensional and dielectric confinement of bound electrons within these naturally occurring quantum-well structures[12], which in turn enhances the joint density of states[26] in the band-overlap integral for $n_2$ nonlinearity. Surprisingly, we find that this effect is pronounced for $n_2$, leaving $\beta$ rather unaffected, seemingly resolving the challenge of the above-mentioned NLO dilemma. Our $(BA)_2(MA)_{n-1}Pb_nI_{3n+1}$ compounds (labeled hereafter by each $n$ value) are, therefore, highly efficient $\chi^{(3)}$ materials with resilience to laser-induced optical damage, which could be vital for advancing multiphoton spectroscopy and microscopy[27, 28], optical switching[29], frequency-resolved optical gating[30], and ultrafast all-optical signal processing for telecommunications[31].

## Results

**Crystal structures of 2D perovskites.** As shown in Fig. 1a, the bandgap-tuning effect is evident in the Ruddlesden-Popper series (Supplementary Note 1). Unlike 3D $MAPbI_3$ ($n = \infty$) that crystallizes in the tetragonal $I4/mcm$ space group at room temperature[32], the 2D perovskites crystallize in the orthorhombic space groups $Pcab$ ($n = 1$), $Ccmm$ ($n = 2$), $Acam$ ($n = 3$), and $Ccmm$ ($n = 4$) (Supplementary Note 2). Each crystal structure consists of finite anionic {$MA_{n-1}Pb_nI_{3n+1}$} perovskite layers, with analogous structural motif to the 3D parent compound (Fig. 1b). The layers expand along the crystallographic $ac$ plane but the growth of the inorganic layers along the $b$ axis is inhibited by the bulky BA cations, thus isolating the perovskite layers from one another. The $c$ and $a$ axes have dimensions related to the $a^*\sqrt{2}$ ordering of the parent perovskite, whereas the dimension along the $b$ axis is ($2a^\star n + x$), where $a^\star = 6.3$ Å is the lattice parameter of $MAPbI_3$, $x \sim 8$ Å is the thickness of the $(BA)_2^{2+}$ bilayer, and $n$ is the number of perovskite layers.

**Nonlinear refractive indices of 2D perovskites.** Figure 2a shows wavelength-dependent third harmonic generation (THG) from $n = 1$–∞ as well as $AgGaSe_2$, when wavelength was varied over a broad range of 1200 nm to 2700 nm. It is interesting to observe fluctuating THG responses of the 2D perovskites, when the THG wavelength, $\lambda_{THG}$, approaches their bandgaps. Detailed THG scans revealed that it arises from resonance effects at both the fundamental gap and the exciton level (Fig. 2b). Considering the narrower bandgap of the 3D perovskite (grey squares), it is expected that $MAPbI_3$ exhibits a stronger THG response than the wider bandgap $AgGaSe_2$ (dots). It is, however, very surprising that the 2D perovskites ($n = 1$–4) having even wider bandgaps yield much stronger THG responses than $AgGaSe_2$ and the 3D perovskite ($n = \infty$) as well, throughout the broad IR regime, i.e., $\lambda > 2100$ nm ($\lambda_{THG} = \lambda/3 > 700$ nm). For example, the $n = 1$–4

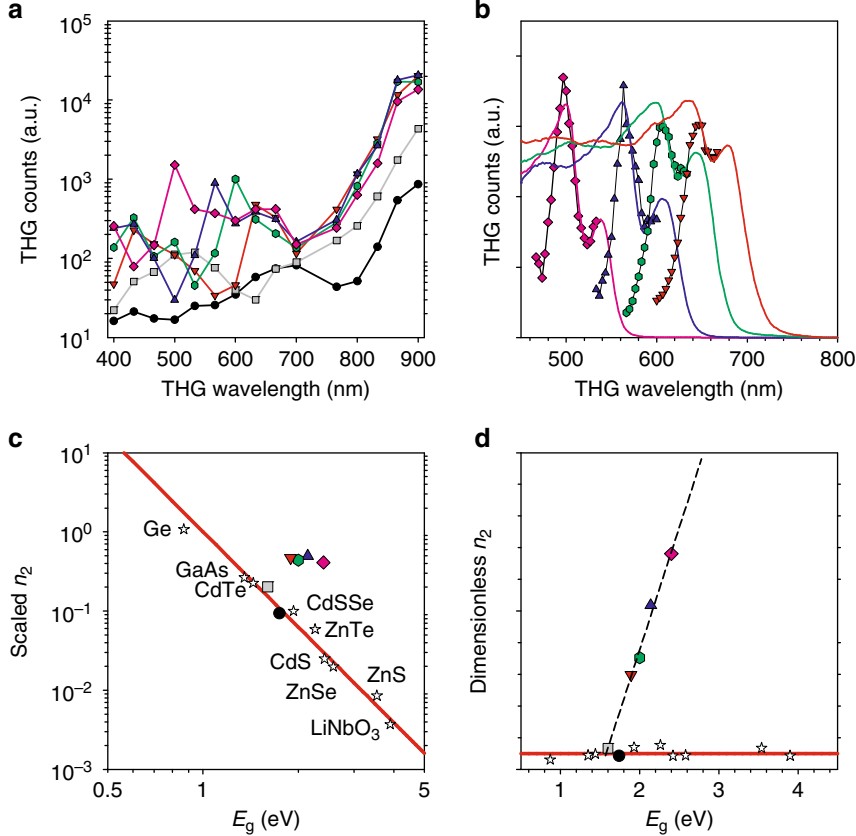

**Fig. 2** Wavelength-dependent third harmonic generation and $n_2$ vs. bandgap. **a** Semi-log plot of broadband third harmonic generation (THG) comparison of $n = 1$ (*purple*), $n = 2$ (*blue*), $n = 3$ (*green*), $n = 4$ (*red*), $n = \infty$ (*grey*), and AgGaSe$_2$ (*black*). Data points are connected by the lines as a guide to the eye. **b** Fine-scale THG spectra scanned across the band edges of the 2D perovskites, overlaid with the measured absorption spectra (*colored traces*). **c** Log-log plot of THG coefficients in terms of the scaled nonlinear refractive index, $80\pi^2 n_2 n_0/(Khc^2\sqrt{E_0}G_2)$, vs. bandgap generated with the experimentally determined $K = 3100$. The *red line* is $E_g^{-4}$. THG coefficients of the 2D perovskites ($n = 1$–4) are very large as evidenced by deviation from the two-band model (*red line*) and exemplary performance in comparison to other inorganic semiconductors (*stars*). **d** Impact of quantum confinement on $n_2$ of the 2D perovskites. The two-band model (*red line*) successfully predicts the actual $n_2$ values of conventional 3D materials (*stars*) as well as the reference 3D materials (*grey square* and *black dot*). However, the strong deviation of the 2D perovskites from the theory is remarkable (*dashed line*), which shows the progressive increase in $\sqrt{E_0}$, which reflects the dipole matrix element between the valence and conduction bands[17]

members of the series outperform the benchmark IR NLO material, AgGaSe$_2$, by more than an order of magnitude even with significant fundamental absorption in the range of $\lambda = 2100$ nm–2500 nm arising from excitation of organic cations (Supplementary Figs. 3 and 4).

We estimated the THG coefficients of $n = 1$–$\infty$ at $\lambda = 2700$ nm, where both the samples and the reference are minimally affected by absorption effects, i.e., at the static limit. By directly comparing with $\chi_R^{(3)} = 1.1 \times 10^{-11}$ esu ($1.6 \times 10^5$ pm$^2$ V$^{-2}$) of AgGaSe$_2$, we calculated $\chi^{(3)}$ values of both 3D and 2D perovskites using[25, 33]

$$\chi^{(3)} = \chi_R^{(3)} \left| \frac{I_S(3\omega)}{I_R(3\omega)} \right|^{1/2}, \quad (1)$$

where $I_S(3\omega)$ and $I_R(3\omega)$ are the THG counts from the sample and the reference, respectively, (Methods and Supplementary Note 3 for negligible effects of the THG coherence length and scattering by powders). Our calculation yields $\chi^{(3)}$ ranging between $2.6 \pm 0.5 \times 10^{-11}$ esu ($n = \infty$) and $5.6 \pm 1.0 \times 10^{-11}$ esu ($n = 2$) with intermediate values of $5.5 \pm 0.9 \times 10^{-11}$ esu ($n = 4$), $5.1 \pm 0.9 \times 10^{-11}$ esu ($n = 3$), and $4.6 \pm 0.7 \times 10^{-11}$ esu ($n = 1$) (Table 1). Note that the $\chi^{(3)}$ values of the 2D Ruddlesden-Popper perovskites are roughly five times larger than that of the reference. This corresponds to 25 times in terms of actual THG counts.

In the static limit, $\chi^{(3)}$ is purely real and represented in terms of $n_2$, which is predicted based on the two-band model[16, 17];

$$n_2(x) = \frac{9.43}{n_0}\chi^{(3)}(x) = K\frac{hc^2}{80\pi^2}\frac{\sqrt{E_0}}{n_0 E_g^4}G_2(x) \text{ in esu} \quad (2)$$

where $h$ and $c$ are the Planck constant and speed of light in vacuum, the Kane parameter, $K$, and $E_0$ ($\sim 21$ eV for typical 3D semiconductors) are constants, and $G_2(x)$ is the $n_2$ dispersion function (Methods) with $x = (hc/\lambda E_g)$ being the dispersion parameter. The large $n_2$ effect is best displayed by plotting the scaled nonlinear refractive indices, $80\pi^2 n_2 n_0/(Khc^2\sqrt{E_0}G_2)$, as a function of $E_g$ on a log-log plot (Fig. 2c). Here, the band dispersion $G_2$ at $\lambda = 2700$ nm was explicitly taken into account and $n_0$ was derived from the Wemple-DiDomenico model for organic–inorganic halide perovskites; $n_0^2 = 1 + 8.32/E_g$ eV [34]. For AgGaSe$_2$, the experimental $n_0$ value [35] was used. $K = 3100$ was determined such that the bulk materials, $n = \infty$ (grey square) and AgGaSe$_2$ (black dot), are fitted by the two-band model (red line), which is simply $E_g^{-4}$. Theoretically the value of $K$ can range from 1940 to 5200 in units such that $n_2$ is in esu [17]. As discussed below, $K = 3100$ is also consistent with the $K$ value independently determined from the 2PA analysis (Fig. 3d). Therefore, $\beta$ and $n_2$ of the 3D materials are consistently explained by the two-band

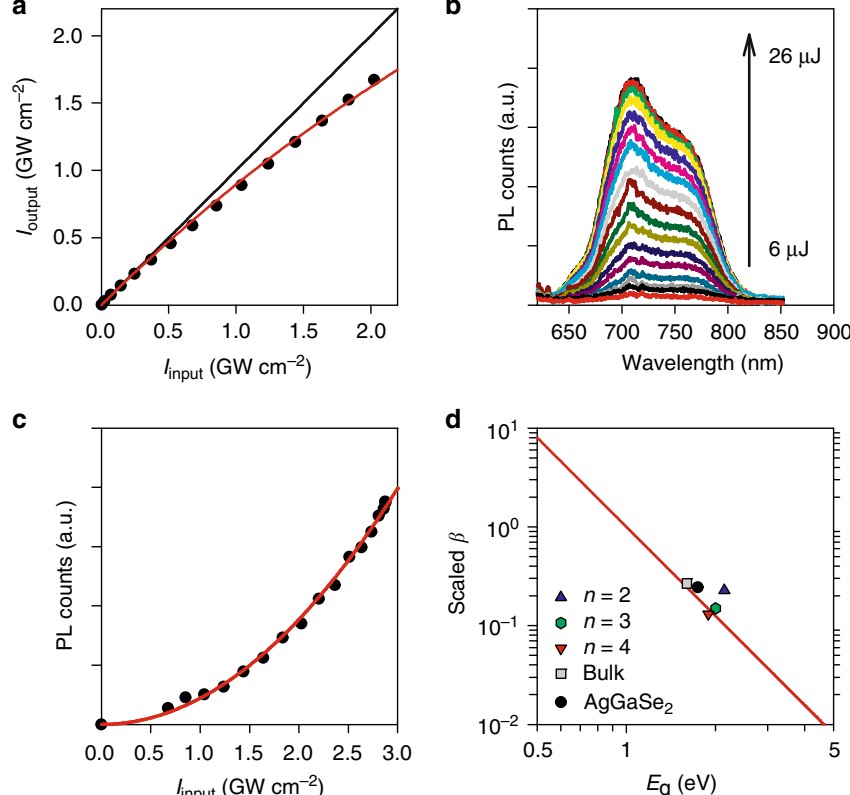

**Fig. 3** Two-photon absorption and photoluminescence characteristics and two-photon absorption coefficient vs. bandgap. **a** $I_{input}$ vs. $I_{output}$ for $n = 4$ (*dots*) at $\lambda = 1064$ nm, superimposed with a two-photon absorption (2PA) fit (*red curve*) with $\beta = 21.9$ cm GW$^{-1}$. The *black line* corresponds to $\beta = 0$. **b** 2PA-induced photoluminescence (PL) spectra ($n = 4$) at $\lambda = 1064$ nm under various excitation levels. **c** Intensity dependence of the resulting PL from $n = 4$ (*dots*) fitted with the square law (*red curve*), indicating the case for 2PA. **d** Log–log plot of the scaled 2PA coefficient, $\beta n_0^2 / (K \sqrt{E_0} F_2)$, vs. bandgap generated with the experimentally determined $K = 5000$. The data point for $n = 1$ is missing because in this case 2PA is purely excitonic and $F_2(x)$ is not well defined. The *red line* is $E_g^{-3}$

**Table 1 The Bandgap energy ($E_g$), exciton-binding energy ($E_{ex}$), third-order susceptibility ($\chi^{(3)} = n_0 n_2 / 9.43$), and two-photon absorption coefficients ($\beta$) of 2D Ruddlesden-Popper perovskites ($n = 1$–4) and 3D materials ($n = \infty$ and AgGaSe$_2$)**

| Material | $E_g$ (eV) | $E_{ex}$ (meV) | $\chi^{(3)}$ ($10^{-11}$ esu) | $\beta$ (cm GW$^{-1}$) |
|---|---|---|---|---|
| $n = \infty$ | 1.60 | $40 \pm 5$ | $2.6 \pm 0.5$ | $46.5 \pm 8.4$ |
| $n = 4$ | 1.89 | $60 \pm 5$ | $5.5 \pm 0.9$ | $21.9 \pm 3.7$ |
| $n = 3$ | 2.00 | $76 \pm 5$ | $5.1 \pm 0.9$ | $20.7 \pm 3.5$ |
| $n = 2$ | 2.14 | $88 \pm 5$ | $5.6 \pm 1.0$ | $18.4 \pm 3.1$ |
| $n = 1$ | 2.40 | $93 \pm 5$ | $4.6 \pm 0.7$ | $15.3 \pm 2.6$ |
| AgGaSe$_2$ | 1.72 | – | 1.1 | $39.9 \pm 8.1$ |

Reducing the dimensionality of the perovskite simultaneously enhances both $\chi^{(3)}$ values and LIDTs, where the latter is evidenced by the decrease of $\beta$

model with the single Kane parameter through the Kramers-Kronig relation.

Our 2D perovskites however defy this universal bandgap scaling as demonstrated by the marked deviation from the red line and exhibit very strong $n_2$ nonlinearity when compared with other notable semiconductors[17, 36] (*stars* in Fig. 2c): Ge has a larger $n_2$ value due to a much narrower bandgap (0.87 eV) but at the expense of a poor LIDT ($\beta \sim 70$ cm GW$^{-1}$ at $\lambda = 2050$ nm using 10-ps pulses)[37]; note that Ge undergoes severe damage via one-photon absorption at typical Nd:YAG radiation ($\lambda = 1064$ nm), which was used for accessing $\beta$ in this study. For instance, the enhancement factor for $n = 1$ (purple diamond) is more than an order of magnitude when compared with the

theoretical prediction. This large nonlinearity apparently arises from 2D confinement that greatly enhances $E_0$ in Eq. (2) despite being scaled down by their relatively wide bandgaps. Figure 2d plots dimensionless $n_2$, which is obtained by multiplying the scaled $n_2$ in Fig. 2c by $E_g^4$, demonstrating the effect of strong confinement without the $E_g^4$ dependence.

**2PA coefficients of 2D perovskites**. This 2D-confinement effect on the other key NLO parameter, LIDT, of the perovskites ($n = 1$–$\infty$) was investigated by measuring $\beta$ values at $\lambda = 1064$ nm using a reflection geometry (Methods); the higher the $\beta$ value, the more susceptible a material is to optical damage via 2PA. Figure 3a plots the sample reflectance of the fundamental beam for $n = 4$, when the input intensity, $I_{input}$, was varied up to 2.2 GW cm$^{-2}$. The solid line represents the case for input = output, where fundamental depletion is absent ($\beta = 0$). The measured reflectance follows this line at lower excitation levels, but deviates gradually from it for $I_{input} > 0.5$ GW cm$^{-2}$, indicating that the fundamental beam is depleted by 2PA. The corresponding $\beta$ value was estimated by fitting Eq. (3) to the observed normalized reflectance, the ratio between output and input:

$$\frac{I_{output}}{I_{input}} = \frac{1}{1 + \beta d I_{input}}, \qquad (3)$$

where $d = 90$–125 µm is the powder size for our reflection geometry[25, 38]. The red trace is a fit generated with $\beta = 21.9 \pm 3.7$ cm GW$^{-1}$, where the uncertainty arises mostly from that of the powder size, $\delta d = \pm 17.5$ µm. The $\beta$ values for the other compounds are $15.3 \pm 2.6$ cm GW$^{-1}$ ($n = 1$),

$18.4 \pm 3.1 \, \mathrm{cm \, GW^{-1}}$ $(n = 2)$, $20.7 \pm 3.5 \, \mathrm{cm \, GW^{-1}}$ $(n = 3)$, and $46.5 \pm 8.4 \, \mathrm{cm \, GW^{-1}}$ $(n = \infty)$, respectively (Supplementary Fig. 5 and Table 1). As a consistency check, we also measured $\beta = 39.9 \pm 8.1 \, \mathrm{cm \, GW^{-1}}$ for AgGaSe$_2$. This $\beta$ value is acceptable within the two-band model, but presumably overestimated by a factor of two when compared with an accurate value obtained from bulk single crystals[39].

At first glance, 2PA of $n = 1$ seems unreasonable considering its wide bandgap of 2.40 eV, which is larger than twice the fundamental energy at $\lambda = 1064$ nm, i.e., $E_g$ $(n = 1) = 2.40 \, \mathrm{eV} > 2$ $(hc/\lambda) = 2.33$ eV. However, 2PA is indeed allowed through a direct excitonic transition located around 2.307 eV ($\sim 540$ nm) as clearly seen in the fine-scale THG and absorption spectra (Fig. 2b and Supplementary Fig. 1e). The exciton-binding energy is, therefore, $\sim 93$ meV at room temperature (Supplementary Fig. 2b), which is significantly larger than $\sim 40$ meV in the bulk[40] (Supplementary Note 1). Thus, $\beta = 15.3 \, \mathrm{cm \, GW^{-1}}$ $(n = 1)$ should be interpreted in terms of resonantly enhanced 2PA at the exciton state[41]. A relatively large $\beta = 18.4 \, \mathrm{cm \, GW^{-1}}$ of $n = 2$ (blue triangle in Fig. 3d and Supplementary Fig. 11) is also affected by similar excitonic effects because of proximity of its optical gap to the 2PA energy (Supplementary Fig. 1d). Nevertheless, the trend of decreasing $\beta$ with lower dimensionality is evident and expected from the theoretical model.

Being excellent light absorbers and emitters under ordinary one-photon absorption (1PA)[11, 12], the 2D perovskites also yield bright PL under 2PA. Figure 3b shows a series of PL spectra observed from $n = 4$ when the incident pulse energy at $\lambda = 1064$ nm was varied from 6 μJ to 26 μJ. The 2PA-induced PL is basically excitonic, but distinct from 1PA-induced PL because of the secondary peak arising from a defect-induced transition. The corresponding intensity dependence agrees well with the 2PA case (Fig. 3c), where the red curve is a power-law fit, $y = ax^b$, with the critical exponent of $b = 2$ (Supplementary Figs. 6 and 7 for other perovskites and AgGaSe$_2$). Our spectroscopic method further reveals the 2PA efficiency and the transition nature of the subsequent PL (Supplementary Figs. 8–11), from which the 2PA dispersion of the perovskites was precisely determined over our broad wavelength range.

The 2PA dispersion based on the two-band model is described by[16, 17]

$$\beta(\lambda, E_g) = K \frac{\sqrt{E_0}}{n_0^2 E_g^3} F_2(x), \qquad (4)$$

where $F_2(x)$ is the $\beta$ dispersion function (Methods). In order to observe the $E_g$ dependence, we plot scaled 2PA coefficients, $\beta n_0^2 / (K \sqrt{E_0} F_2)$, in Fig. 3d on a log-log scale. The band dispersion was taken into account through each $F_2$ value. Note that a single parameter of $K = 5000$ was consistently used for overall scaling of the data points. As discussed above, this $K$ value agrees well with that determined for $n_2$ ($K = 3100$) in light of the systematic overestimation of $\beta$ by a factor of two. The red line in Fig. 3d is, therefore, $E_g^{-3}$ and indicates that the 2PA efficiency of all the perovskites is reasonably well explained by a universal relation without any notable 2D effect for $n = 1$–4 (Supplementary Fig. 11). This result implies that the associated LIDT is enhanced in the lower $n$ (larger $E_g$) compounds in this Ruddlesden-Popper series (Supplementary Fig. 5). This is a very intriguing observation because 2D confinement typically affects both $n_2$ and $\beta$ by the $\sqrt{E_0}$ dependence in Eqs. (2) and (4), which is the essence of the Kramers-Kronig relation. In fact, the enhancement of 2PA is rather common in low-dimensional quantum structures[42], although this is not the case for $n = 1$–4. Therefore, our experimental result of selective enhancement of $n_2$ without any noticeable increase in $\beta$ is quite exceptional, where the latter effect leads to the enhancement of the LIDT upon bandgap blueshift.

**Relaxation dynamics of excitonic matter under 2PA**. Figure 4a shows the measured PL spectra for $n = 4$ as wavelength was tuned from 1000 nm to 1700 nm at increments of 100 nm. Upon wavelength tuning toward longer wavelengths, several striking observations were made. Firstly, the PL shape progressively evolves from a broad PL band to a low-energy shoulder peaked around 780 nm. The latter stands alone for $\lambda \geq 1400$ nm with a minor contribution from the main exciton PL (solid trace), which was obtained by subtracting the PL at $\lambda = 1200$ nm with 80% scaling from that at $\lambda = 1000$ nm. This exciton PL is best observed under 1PA (dashed trace). Secondly, although relatively weak, 2PA is still allowed even when wavelength is tuned below the exciton resonance ($\lambda \geq 1400$ nm) as evidenced by the shoulder peak (700 nm–850 nm). We confirmed that this feature exists for $n = 3$ and $n = 4$ only (Supplementary Fig. 10). Lastly, this defect-induced PL emission exhibits amplified spontaneous emission (ASE)[43] heralded by the emergence of a superlinearly increasing sharp peak ($\sim 790$ nm) as discovered from power dependence (Fig. 4b). The corresponding onset threshold intensity is about

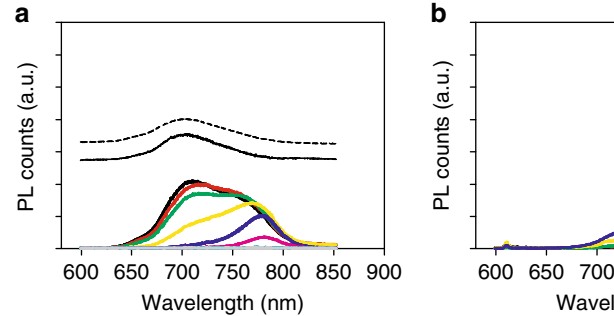
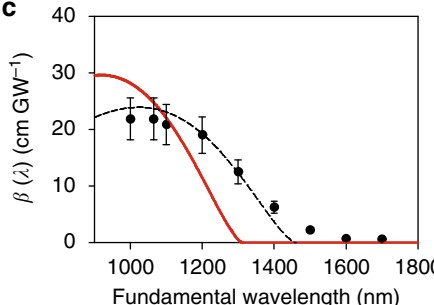

**Fig. 4** Two-photon-absorption-induced photoluminescence and two-photon absorption dispersion of $n = 4$. **a** PL spectra from $n = 4$ under two-photon absorption (2PA) for different excitation wavelengths from 1.0 μm (*black*) to 1.7 μm (*grey*) at increments of 0.1 μm (ascending wavelengths correspond to the color sequence *black, red, green, yellow, blue, purple, cyan, grey*). The major exciton PL (vertically translated for clarity) is identified by spectral subtraction, which is essentially the same as the one-photon-absorption-induced photoluminescence (PL, *dashed trace*) using frequency-tripled Nd:YAG radiation (355 nm). **b** Amplified spontaneous emission of the sharp peak ($\sim 790$ nm) from the low-energy shoulder of $n = 4$ when the exciton state is resonantly excited by 2PA with excitation energy ranging from 10 μJ (*black*) to 40 μJ (*blue*) at increments of 10 μJ (ascending excitation energies correspond to the color sequence *black, red, green, yellow, blue*). **c** Experimental $\beta(\lambda)$ (*dots*) where *error bars* arise from uncertainty in the powder size. The theoretical $\beta(\lambda)$ (*red curve*) corresponds to Eq. (4) using $K = 5000$ and the fundamental bandgap, $E_g = 1.89$ eV. The *dashed curve* is Eq. (4) using the optical gap ($\sim 1.70$ eV) as a band-dispersion parameter with $K = 3500$

1.6 GW cm$^{-2}$. Interestingly, ASE occurs only when the input wavelength is 2PA resonant with the exciton line ($\sim 690$ nm $= \lambda/2 = 1380/2$ nm), which directly populates cold excitons that subsequently decay at defect sites. We confirmed that such ASE is entirely absent for band-to-band 2PA ($\lambda < 1400$ nm) or 1PA, which initially populate hot carriers that are less influenced by defects. This intense and narrow emission at room temperature can be of significant interest for optoelectronic applications.

The 2PA dispersion, $\beta(\lambda)$, within our experimental range was obtained by monitoring the relative PL counts[8, 44]. Figure 4c shows the spectrally integrated PL counts (dots) as a function of wavelength obtained from Fig. 4a, which were scaled in accordance with the absolute $\beta$ value determined at $\lambda = 1064$ nm (Fig. 3a). Clearly, each data point is proportional to the number of carriers generated by 2PA for a given wavelength, which in turn reflects $\beta(\lambda)$[44]. The red curve is the theoretical two-band model, generated with the predetermined value of $K = 5000$ in Eq. (4). Because of significant 2PA below the fundamental gap, the conventional model does not explain 2PA at exciton and defect states. We found that using an optical gap defined by the low-energy onset of excitonic transition better explains the overall 2PA dispersion as indicated by the dashed curve; $K = 3500$ was used to fit the overall $\beta(\lambda)$ (Supplementary Fig. 9) for the other $n$ members).

## Discussion

Our experimental data demonstrated that the layered 2D organic–inorganic iodide perovskites ($n = 1$–4) exhibit very large THG response in the mid-IR regime as clearly evidenced by their $n_2$ values that surpass the high-performance NLO material, AgGaSe$_2$. The absolute 2PA dispersion, $\beta(\lambda)$, of the 2D perovskites can be explained by the two-band model with a nominal Kane parameter. However, the corresponding Kramers-Kronig conjugate is considerably enhanced by the 2D effects. This observation draws a sharp contrast to the benchmark 3D perovskite ($n = \infty$) whose real and imaginary parts of $\chi^{(3)}$ are consistently explained by the existing theory. We emphasize that dimensional reduction significantly scales up $n_2$ of the perovskites over the entire frequency spectrum by quantum confinement. This 2D effect is distinct from resonant NLO effects in 3D perovskites, which are sharp at that particular resonance only[9, 10]; therefore, their high-performance range is severely restricted. We also note that our 2D perovskites are much more practical than highly nonlinear, atomically thin transition metal dichalcogenides, which are only useable in a liquid suspension form[45]. In contrast, our hybrid 2D perovskites can be readily prepared into films[11, 13] where the effect of self focusing can be accumulated over the macroscopic sample thickness when electromagnetic radiation propagates. The 2D perovskites could be especially promising for biological applications working at the mid-IR (Supplementary Fig. 3b) where they exhibit strongest THG responses and are resilient to optical damage at the same time.

The observation of unusual optical nonlinearity may indicate that the simple two-band model is not appropriate to explain our novel hybrid perovskites crystallized in the 2D layered structures. Although it deserves more theoretical investigation on the rigorous treatment of the Kramers-Kronig relation by directly incorporating the band structures of the 2D perovskites with a possible polarization dependence, such a striking NLO effect in our hybrid materials is quite unique and can be further exploited to form a basis for a future design concept of new NLO materials that overcome the long-standing issue between nonlinearity and LIDT, together with the added advantages of cost-effective fabrication. Based on the consistent measurements on the two batches of samples prepared over the extended time frame, we have

confirmed that the primary NLO properties of the 2D perovskite samples persist over 6 months upon sealing (Methods). Finally, highly luminescent PL response by 2PA at room temperature could also be useful for imaging applications.

## Methods

**Sample preparation.** Our perovskite compounds were synthesized from an off-stoichiometric reaction of PbO, MACl, and $n$-BA[12]. All chemicals were purchased from Sigma-Aldrich and used as received. PbO powder was dissolved in a mixture of aqueous HI and aqueous H$_3$PO$_2$ by heating to boiling under constant magnetic stirring for $\sim 5$ min. Subsequent addition of MACl into the hot solution initially caused the formation of a black powder, which rapidly redissolved under stirring to afford a clear bright yellow solution. For the synthesis of $n = \infty$, the stirring of the solution was simply discontinued and the solution was left to cool to room temperature for crystallization of the 3D perovskite. For the synthesis of $n = 1$–4, according to the desired layer number, $n$-BA was added to the bright yellow solution and then subsequently dissolved under heating of the combined solution to boiling. The stirring was then discontinued and the solution was left to cool to room temperature for crystallization of each 2D perovkite compound. All the perovskite samples were isolated by suction filtration and thoroughly dried under reduced pressure. For room-temperature NLO measurements described here, these perovskites were crushed into powder in the range of 90 µm–125 µm after sieving. The samples were placed into borosilicate capillary tubes under a dry nitrogen environment, and then sealed to prevent exposure to moisture and oxygen during measurement. Each capillary tube was loaded into a homemade sample holder that was mounted on a Z-scan translation stage. The optical quality reference powder, AgGaSe$_2$, was also prepared in a similar fashion[38].

**Excitation condition for NLO experiments.** Our NLO experiments on powders of the perovskites and a reference IR NLO material, AgGaSe$_2$, were conducted at room temperature using wavelength-dependent Z-scan nonlinear spectroscopy (WDZNS)[44]. WDZNS is ideal for investigating frequency-dependent $\chi^{(3)}$ effects as a function of input intensity and wavelength. The train of 30 ps fundamental pulses over a broad wavelength range ($\lambda = 1000$ nm–2700 nm) was produced from an optical parametric oscillator (OPO), which was synchronously pumped by the frequency-tripled output (355 nm) of an Nd:YAG laser. The incident pulse energy (20 µJ) from the OPO was adjusted using a combination of a half-wave plate (HWP) and a linear polarizer (LP) before being mildly focused onto the sample using a positive lens ($f = 7.5$ cm) with a spot size of $\sim 140$ µm in Gaussian width. The corresponding sample position was far away from the Z-scan focus ($Z = 0.65$ cm behind the Z-scan focus). This scheme is to minimize the change in the spot size when we vary wavelength over a broad range; the beam waist $w_0$ at the Z-scan focus undergoes a significant wavelength-dependent variation via $w_0 = (\lambda/\pi)(f/\sigma)$, where $f$ and $\sigma$ are the focal length and the Gaussian width of the incident beam, respectively. Any thermal load on the samples by the laser pulses with photon energies tuned below the bandgap was negligible within our experimental intensity range up to $\sim 5.5$ GW cm$^{-2}$ due to the slow repetition rate of 50 Hz.

**THG experiments.** Our wavelength-dependent THG measurements on the perovskite compounds were conducted in the range of $\lambda = 1200$ nm–2700 nm at increments of 100 nm. The corresponding THG wavelength, therefore, ranges from 400 nm–900 nm. Moreover, we conducted fine-scale THG scans across the bandgaps of the 2D perovskites to probe the resonance effect (Fig. 2b). The THG signal from the sample passed through a combination of collection lenses and was guided via fiber optics into a selective-grating spectrometer coupled to a charge couple device (CCD) camera. The relative THG signals recorded in a broad wavelength range were precisely calibrated with the known and measured efficiencies of all optical components. We confirmed that second harmonic generation (SHG) and any THG from other optical components are negligible. The absolute THG coefficients of the perovskites ($n = 1$–$\infty$) at the static limit ($\lambda = 2700$ nm) were estimated by directly comparing with that of the reference material, AgGaSe$_2$, of an optical quality prepared with a similar fashion (Supplementary Fig. 4); $\chi_R^{(3)} = 1.6 \times 10^5$ pm$^2$ V$^{-2}$ ($1.1 \times 10^{-11}$ esu)[33]. As a noncentrosymmetric material, AgGaSe$_2$ exhibits highly efficient SHG. It also yields 2PA-induced PL at room temperature. Because of a large THG phase mismatch, the corresponding coherence length was not experimentally accessible, which is smaller than the minimum powder size: For example, the theoretical THG coherence length of AgGaSe$_2$ is about 4 µm at 2700 nm[35]. However, we can assume that the THG coherence lengths of the samples are comparable to the reference because of their similar indices of refraction especially at the long-wavelength limit, where $\chi^{(3)}$ is purely real without any abrupt change arising from resonance effects.

**2PA experiments.** To determine the 2PA coefficient, $\beta$, absorption of fundamental Nd:YAG radiation (1064 nm) by the sample was monitored as a function of input intensity using the same collection optics for the THG experiment under far-field reflection geometry. Since our samples are powder, a typical open-aperture Z-scan was not accomplished. Instead, the input intensity, $I(\phi)$, was tuned over two orders of magnitude by the combination of the HWP and the LP at a fixed $Z = 0.65$ cm

behind the focus, where $\phi$ is the HWP angle. First, the fundamental input intensity, $I_b(\phi)$, via sample reflection was measured as a function of $\phi$ with a 5% neutral density (ND) filter before the sample, thereby ensuring no 2PA. This linear reflectance can be used as a reference for generating the normalized reflectance. With the ND filter placed in the reflection path after the sample, the sample reflectance in the nonlinear regime, $I_a(\phi)$, was measured to determine $\beta$, where we gradually varied $\phi$ starting from the lowest intensity. At low excitation levels, $I_a(\phi)$ $= I_b(\phi)$ is expected. However, efficient 2PA would cause a gradual reflection loss in $I_a(\phi)$ when the input intensity increases. In other words, the ratio $I_a(\phi)/I_b(\phi)$ corresponds to the normalized sample reflectance. Therefore, $I_b(\phi)$ and $I_a(\phi)$ correspond to the input and output intensities, respectively, in Fig. 3a and Supplementary Fig. 5. This method was utilized for characterizing three-photon absorption in a single crystalline $CsPbBr_3$ under transmission geometry[8]. The perovskites yield strong 2PA-induced PL emission. The wavelength-dependent 2PA efficiency was, therefore, compared by monitoring the resulting PL brightness as wavelength was tuned within the 2PA band and below due to excitonic and defect transitions with the intensity kept constant. The PL method presented here also relies on the special capability of WDZNS[44], which can spectroscopically resolve the PL signal. Combined with the absolute measurement at 1064 nm, wavelength-dependent PL measurement at the same fixed Z completely determines the absolute 2PA dispersion, $\beta(\lambda)$.

**Dispersion functions for nonlinear refractive index and 2PA.** $G_2(x)$ and $F_2(x)$ for $n_2$ and $\beta$ relate to the real and imaginary parts of third-order nonlinearity. They can be calculated using a perturbation theory. Assuming two parabolic bands for optical transitions, they are given by[16]

$$G_2(x) = \frac{1}{(2x)^6}\left\{4\left[1-(1-x)^{3/2}-(1+x)^{3/2}\right] - \frac{3}{4}x^2\left[(1-x)^{-1/2}+(1+x)^{-1/2}\right]\right.$$
$$+ 6x\left\{(1-x)^{1/2}-(1+x)^{1/2}\right\} + 2\left[H(1-2x)(1-2x)^{3/2}+(1+2x)^{3/2}\right]\right\}$$
$$+ \frac{1}{(2^{10}x^5)}\left\{70x^2+3x\left[(1-x)^{-1/2}-(1+x)^{-1/2}\right]\right.$$
$$\left.\left. - \frac{1}{2}x^2\left[(1-x)^{-3/2}+(1+x)^{-3/2}\right]\right\}\right.,$$

$$(5)$$

$$F_2(x) = \frac{(2x-1)^{3/2}}{(2x)^5}H(2x-1),$$

$$(6)$$

where $H(x)$ is the Heaviside step function.

**Data availability**. The experimental data that support the findings of this study are available from the corresponding author upon request.

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

## Acknowledgements

This work was supported by Basic Science Research Program (2017R1D1A1B03035539) through the National Research Foundation of Korea (NRF), funded by the Korean government. C.C.S. and M.G.K. acknowledge the support under ONR Grant N00014-17-1-2231. We thank Gooch and Housego for generously supplying a high-quality $AgGaSe_2$ crystal.

## Author contributions

J.I.J. initiated and directed the study. F.O.S. and J.W. carried out nonlinear optical experiments and analyses. C.C.S. and M.G.K. prepared perovskite samples, conducted basic materials characterization, and contributed to writing the manuscript. All authors discussed the results and wrote the manuscript.

## Additional information

**Competing interests:** The authors declare no competing financial interests.

**Change history:** A correction to this article has been published and is linked from the HTML version of this paper.

