## [Peer Review File · Nature Communications]

Editorial Note: Parts of this peer review file have been redacted as indicated to remove third-party material where no permission to publish could be obtained.

Reviewers' comments:

Reviewer #1 (Remarks to the Author):

I am writing concerning the article entitled "Selective enhancement of optical nonlinearity in two-dimensional organic-inorganic lead iodide perovskites", by F. O. Saouma, C. C. Stoumpos, J. Wong, M. G. Kanatzidis, and J. I. Jang.

The authors investigate the third order optical nonlinearities of a series of hybrid lead iodide perovskites and the nonlinear refractive index (real part of the third order nonlinear susceptibilities) are determined. They show that a high broadband nonlinear optical efficiency is exhibited by these systems due to a dimensional reduction resulting to a dielectric confinement. Additionally the possibility to tailor this type of materials in order to enhance both the nonlinearity and the laser-induced damage threshold is discussed and details are given. This is of great importance for photonic applications where high laser damage thresholds are required. Additionally, as stated in the article, the presence of two photon absorption can influence the performance of the systems, both by influencing their general attribute for photonic applications, such as optical switching, as well as their laser damage thresholds. Indeed it is well known that a significant decrease of the laser damage thresholds can occur by an increase of the nonlinear absorption of the systems. For this reason an additional detailed study of the two photon absorption of the systems is performed.

The article is well and clearly written and taking into account the importance of such nonlinear optical systems for photonic applications it can be of interest for the readership of Nature Communications. A weak point concerns the structure of the article. The reader who wants to get into the details of the work has to spend a lot of time searching in the text or in the supplementary information. The authors have to introduce some important information inside the text and facilitate the reader with additional tables, when possible.

Below a list with some important points to be taken into account are shown:

- a) A general table should be given presenting in details the nonlinearities of all the members of the series investigated. It will provide the reader with a quick reference and comparison between the systems. It will also indicate the difference between the real and imaginary parts of the third order nonlinear susceptibilities.
- b) It is important that the main experimental details of the third harmonic generation studies are introduced in the text.
- c) In several cases throughout the main text, the laser damage thresholds are compared with previously studies systems, which without doubt is an important aspect of the work. However important parameters, like the pulse duration regime and the irradiation wavelengths are not given. The authors should detail more and explain under which excitation parameters the investigated series can provide a benefit with comparison to previously investigated systems with significant performances.
- d) The authors comment the relation of the nonlinear absorption parameters with the dimensionality of the system. In this case more details should be given to improve the understanding, which is not straightforward.
- e) In the S3.1 paragraph the wavelength dependence of the third harmonic generation part of the work is detailed. The laser sources shall be cited in this case and some more details concerning the wavelength-dependent Z-scan nonlinear spectroscopy shall be added.

After these additions the article can be re-considered for publication in Nature Communications Journal.

Reviewer #2 (Remarks to the Author):

The work 'Selective enhancement of optical nonlinearity in two-dimensional organic-inorganic lead

iodide perovskites', authored by F. O. Saouma, C. C. Stoumpos, J. Wong, M. G. Kanatzidis, and J. I. Jang is very interesting and well written. They present exciting data on nonlinear optical (NLO) properties of hybrid perovskites of reduced dimensionality. In particular they show that that dimensional reduction of the perovskite material enhances the broadband NLO efficiency, decreasing the damage threshold.

Their data are consistent, and their interpretation well augmented.

My main concern regards the following few points:

- Why the measurements are not performed on single crystals (as expected for NLO experiments) of each perovskite and on powder instead? Please discuss this point. Please add details in SI on how you performed the measurements on the powders.
- What is the contribution of scattering from the sample powders on the measurement ?
- Even if it clear that in the lower dimensionality perovskites the air stability behaviour is improved respect to the standard ($n=\infty$) one, it is still a main weakness of these material. While any materials to be suitable for NLO applications, It needs have stable performances in air for prolonged times, also under illuminations. Have the performances been tested in days' time scale? In which conditions of Humidity? Please provide details on these critical points

In terms of results' presentation I have found the figures not extremely clear. Moreover there are many interesting data in SI, that it would be better to include in the main manuscript that has just 3 figures now. So I suggest including some of the figures in the main text, as they can help the reader to follow the paper. I also suggest some change on the manuscript, listed here after.

Specifically my suggestions are;

-Modify the symbols in figure 1 since they are not very clear to distinguish among the different n values when printed in back an white. Increase figure 1 a quality

-Include figure S3b in the main text

-On pag. 8 of the main manuscript please include the excitation binding energy values for $n=\infty$, for a direct comparison

-On pag 9, second paragraph please invert the discussion of figure 3b, discussing the main figure first and then the inset. Please comment the power law dependence and the exponent value.

-It would be good to include figure 9 c in the main text to support the description of the 2PA phenomenology

-I will expand the asset sentence of the conclusions, highlighting he importance of using infrared excitation for biological application.

Reviewer #3 (Remarks to the Author):

The paper submitted by Jang et al. demonstrate the enhancement of optical nonlinearity in two-dimensional organic-inorganic lead iodide perovskites. It is an interesting study with their measurements referenced to conventional AgGaSe powder concurrently to demonstrate how this 2D perovskites different from the conventional one. They attribute this to the quantum confinement based on the parameters of their fitting. However, I do not find the explanation of such enhancement of optical nonlinearity in these materials convincing. The materials seem to have considerable subgap states, in-particular the low 'n' perovskite. There is no further theoretical approaches provided by the authors to relate how the quantum confinement leads to the enhancement. As a result, I cannot support the publication of this article to Nature Comm. I have some doubts about some technical points. The author claims the exciton binding energy of MAPbI3 ($n = \infty$) is around 40 meV, which is in-marked contrast with the general observation of few meV of these materials and that the optical absorption is dominated by band-to-band transition, which also appears in their measurements. These measurements are conducted with powder form. To obtained the X(3) of the materials, the authors relate the parameter to the AgGaSe. I am not sure how accurate such measurements are, for example, whether the interparticle scatter within the powder, self-absorption have been taken into account. The authors should also provide the absorption spectra that covered at least 2700nm. The materials used in

this study may have considerably subgap states. From the resonant 2PA PL, it is also clear that the emission linewidth is considerably broader as n is smaller. It is not clear the effects of these subgap states and I suspect that the larger LIDT observed in smaller n may be due to the presence of such subgap states. Should the author improve the quality of the materials, the output could be different. The 2PA measurements shown in Figure S5 used excitation wavelength of 1064nm. This seems alright for large n perovskites but the 2 photons energy is below that of bandgap small n perovskite and that the subgap states appears in low n perovskites could play a dominant role in the interpretation. Should the authors provide more measurements on sample with negligible subgap emission, the value of beta might be different.

Response to the referees' report

Reviewer 1's Comments

- 1) A table should be given summarizing the nonlinearities of all the members for quick comparison for the reader.
- 2) Experimental details on third harmonic generation (THG) should be described in the text, not in Supplementary Information (SI).
- 3) Important parameters, like the pulse duration regime and the irradiation wavelength should be given for the previously studied systems. The authors should explain under which excitation parameters the investigated series can provide a benefit with comparison to previously investigated systems with significant performance.
- 4) More details can be given to improve the understanding of the relation between nonlinear absorption parameters and the dimensionality of the system.
- 5) In S3.1, the laser sources shall be cited and some details on the wavelength-dependent Z-scan nonlinear spectroscopy shall be added.

Author's Response

- 1) We appreciate the reviewer's suggestion and have tabulated the measured nonlinear optical (NLO) parameters in Table 1.
- 2) Experimental details for THG as well as 2PA now appear in Methods of the main manuscript.
- 3) We have provided the experimental parameters for Ge (ref 37) and clearly indicated its limitation under Nd:YAG illumination in the text. In Discussion, we have also added a sentence, highlighting the benefit of the 2D perovskites with performance specification.
- 4) Besides the two-band model already described in our manuscript, at this time, it is difficult to expound the underlying theory that explains the relation between nonlinear absorption and the dimensionality. Our study is an "experimental" body of work that clearly demonstrates the current limitation of the simple model for the emerging 2D perovskite system and we do hope that our results will initiate the detailed calculation of the associated matrix elements by properly accounting for the band structures of these 2D materials. We believe that the reviewer would understand this point.
- 5) We have revised the manuscript such that experimental details appear at the main text (Methods).

Reviewer 2's Comments

- 1) The authors should explain why powder samples were used rather than single crystals. Also some details regarding powder measurements should be given in SI.
- 2) Any scattering effect arising from the powder measurement should be described.

- 3) The weakness of the perovskites might be instability. The authors should provide details on the performance of the perovskites upon exposure to the ambient condition or over an extended time.
- 4) Just in terms of presentation, the authors can consider the specific suggestions of the reviewer;
 - Improve the quality of figure 1.
 - Include figure S3b in the main text.
 - Include the exciton binding energy value for $n = \infty$ in page 8.
 - Reorganize the paragraph that discusses figure 3b and comment on the power law behaviour in page 9.
 - Include figure S9c in the main text.
 - Include a sentence that highlights the importance of using infrared excitation for biological applications.

Author's Response

- 1) We understand the point of the reviewer because it would be ideal to conduct measurements on single crystals. However, at this point, single crystals with a “macroscopic size” are not available for measurements, especially polarization dependence. In Supplementary Section 3, we have provided some details on the Kurtz-Perry powder method that is quite reliable for the early assessment of a new NLO material. We believe that the reviewer would understand the current limitation of our work.
- 2) We appreciate the reviewer for pointing out this issue. In Supplementary Section 3, we have detailed why any effect arising from scattering by powders is negligible for our specific experimental conditions in terms of reflection geometry and powder particle size in the Mie scattering regime.
- 3) In general, all hybrid halide perovskites have a temporal instability issue. Nonetheless, in our previous work (ref 11), we showed that our 2D perovskites significantly outperform the 3D counterpart in terms of their robustness during two months of testing under a 40% humidity level. In fact, we also confirmed that the perovskites can persist over at least 6 months upon sealing. In Discussion, we have added a relevant sentence to highlight the stability of our 2D materials.
- 4) We have edited the manuscript according to all the useful suggestions of the reviewer.

Reviewer 3's Comments

- 1) The authors should provide further theoretical approaches to how confinement leads to the enhancement of nonlinearity.
- 2) The authors should discuss why the exciton binding energy for $n = \infty$ (~40 meV) is different from some measured values of few meV reported elsewhere.

- 3) The authors should provide some details on the powder technique for establishing the accuracy of the measurements, ensuring no contribution from scattering and self-absorption effects.
- 4) The authors should provide the absorption spectra that covers up to 2700 nm at least.
- 5) The larger laser-induced damage threshold (LIDT) in smaller n values may be due to the presence of subgap states. If possible, the effect of subgap states on two-photon absorption (2PA) can be tested in the subgap-state-free $n = 1$ sample, in which the 2PA coefficient can be different.

Author's Response

- 1) Please see our response to the fourth issue of Reviewer 1.
- 2) Because of exciton-phonon coupling, determining the exciton binding energy of MAPbI₃ is known to be rather tricky in deconvoluting the exciton peak when using spectroscopic methods, even at cryogenic temperatures. Indeed, there are widely varying values in the literature depending on the sample quality and measurement techniques employed. Given this said, a finite exciton binding energy from our measurements does not contradict typical values determined based on “spectroscopic” methods. Most of all, we emphasize that a small change in the energy from 0 to 40 meV does not affect the main plots of nonlinearity vs. bandgap for the bulk perovskite. To clarify the issue, we have added relevant sentences, modified Fig. S2, and newly introduced ref 4 in SI, which seriously discuss the exciton binding energy in 3D perovskites.
- 3) We appreciate the reviewer for pointing out this issue. In Supplementary Section 3, we have provided some information on the Kurtz-Perry powder method and detailed why any effect arising from scattering by powders is negligible for our specific experimental conditions in terms of reflection geometry and powder size in the Mie scattering regime. The effect of self-absorption is absent because the THG coefficients were evaluated in the static regime where THG signal is not reabsorbed by the sample (Fig. S4).
- 4) We have provided FTIR spectra of the perovskites in SI (Fig. S3b). The observed peaks are all intrinsic to the hybrid perovskites, which are not subgap states. As mentioned in the corresponding figure caption, the minor absorption at 2700 nm hardly affects our THG coefficients from which we determined the nonlinear refractive indices. In fact, this absorption effect tends to weaken the THG response (see the THG counts at 700 nm–800 nm range in Fig. S4). This implies that the actual THG coefficients can be even higher.
- 5) We agree that our perovskite samples have some subgap states. However, our manuscript (in pages 8–9) clearly indicates that 2PA in $n = 1$ arises from “intrinsic” excitonic transition when excited at 1064 nm, which is not associated with any extrinsic subgap-induced transition. This is also clearly evidenced by Fig. S1e, showing the exciton peak whose spectral location overlaps with the two-photon energy of Nd:YAG radiation. We emphasize that both the real and imaginary parts of optical nonlinearities

of our samples were determined by carefully selecting the input wavelengths in which subgap-related phenomena are absent or minimal.

Reviewers' comments:

Reviewer #1 (Remarks to the Author):

The authors improved the article taking into account the comments of the reviewers. The article can be accepted for publication in nature communications in its present form.

Reviewer #2 (Remarks to the Author):

The manuscript 'Selective enhancement of optical nonlinearity in two-dimensional organic-inorganic lead iodide perovskites' has significantly improved. The data are now presented in a more clear way. The manuscript is now easier to be read and understood. I am happy with the changes and the manuscript should be accepted for publication at this stage.

Reviewer #3 (Remarks to the Author):

The authors have revised the manuscript. From the revised manuscript with added description on their experiments, the authors have answered most of my questions. However, it will be good if the authors can provide the measurements as well done on thin-film, where the scattering effects from the particles can be ignored without any assumption. Their samples look rather inhomogenous based on both absorption and emission spectra. It casts the doubt as well the scattering effects complicated by the particle size, grain boundaries effect and etc. I am curious why this is not done during the revision.

It is true that the theoretical treatment may be premature at this stage. However, that actually rises the doubts if the 2-band model used to calculate the n_2 is meaningful or not. The manuscript used the conventional model to predict the 'unconventional' crystal sounds unconvincing to me, at least it casts the doubt about the derived n_2 parameter for the layered 2D perovskites from the experimental measurements. It could also mean that the model isn't appropriate to extract n_2 without further justification and that the selective enhancement of n_2 but not β wrt E_g isn't convincing enough.

Accordingly, I'm hesitant to recommend publication based on current stage.

Response to the third referee's report

The other two reviewers have suggested the publication of the manuscript after the first revision.

Reviewer 3's Comments

- 1) It would be good if the authors can provide the measurements on thin films because scattering effects complicated by the particle size and grain boundaries can be ignored without any assumption.
- 2) The powder samples seem rather inhomogeneous based on both absorption and emission spectra, which may arise from scattering effects.
- 3) Calculation of n_2 based on the two-band model seems not meaningful, because this conventional model may not predict/extract optical nonlinearity of the unconventional 2D layered materials appropriately.

Author's Response

- 1) While thin-film measurement may reduce minor scattering effects, it can lead to even more inaccurate estimation of optical nonlinearity because of the formation of "submicron-size" grains typically present in perovskite films; see for example Fig. 1. It is well known that such fine grains in a thin film causes significant enhancement in the nonlinear optical response due to extra charges accumulated on the grain boundaries; see for example, Appl. Phys. Lett. 73, 572 (1998) and Opt. Express 13, 9211 (2005) for ZnO films.

Fig. 1 | SEM image of a thin film derived from our 2D perovskite: Image from our previous publication; Nature 536, 312 (2016).

[Redacted]

Fig. 2 | SEM image of powders: Image from our previous publication; Chem. Mater. 28, 2852 (2016). Scale bar = 200 microns.

In contrast, our powders are crystallites with a much larger grain size as shown in Fig.2. This means that measuring nonlinearity using thin films is not ideal. For instance, n_2 values measured “below” the bandgap of the perovskite films are 10^{-12} cm²/W [MAPbBr₂I: J. Mater. Chem. C 4, 4847 (2016)], 36×10^{-12} cm²/W [MAPbI₃: ACS Photon. 3, 361 (2016) & ACS Photon. 3, 371 (2016)], and 84×10^{-12} cm²/W [MAPbI_{3-x}Cl_x: ACS Photon. 3, 371 (2016)], respectively. Among these widely varying values, only 10^{-12} cm²/W corresponds to the off-resonant case: The other cases are strongly enhanced by subgap-state resonance. But this value of 10^{-12} cm²/W is still likely overestimated by the impact of grain boundaries.

In fact, our measured n_2 value of MAPbI₃ powder is about 0.2×10^{-12} cm²/W (2.6×10^{-11} esu) at the off-resonant static limit. Considering the issue of grain boundaries causing a larger value of 10^{-12} cm²/W in the thin film, our powder value is quite reasonable and indeed in excellent agreement with the two-band model; note here that the model is good for 3D perovskites; see also ACS Nano 9, 9340 (2015) for the validity of the model for the 3D perovskite, MAPbBr₃. Moreover, the accuracy of our powder technique is already established within a “factor of 2” using a benchmark reference material (AgGaSe₂). In Supplementary Note 3, we have delineated our carefully designed experimental scheme (Mie scattering regime in reflection geometry) that minimizes the effects of scattering.

Therefore, we do not think that it is particularly meaningful to prepare thin films and conduct the series of experiments on them especially at this extra revision round; we strongly believe that our manuscript is pretty mature for publication as also recommended by the other two reviewers.

- 2) If “inhomogeneity” by the reviewer means that our samples contain some levels of subgap states, we totally agree: The subgap states are evident from our 2PA-induced PL. But we do not think that our samples are inhomogeneous (except for random orientation of powders), because we did not see any difference in the absorption and PL spectra when we scanned the excitation spot over the various sample area. This is described in Supplementary Note 3. Most importantly, we already addressed in the first revision that the presence of subgap states does not alter our main conclusion. Also, the impact of the random nature of powders on the measured absorption and PL spectra is not critical in line with our response to the first issue above.

- 3) We appreciate the reviewer for pointing out this possibility. Yes, we agree that the simple model may not be appropriate for explaining unusual nonlinearity of unconventional materials. In the revised manuscript, we have added a sentence that clearly mentions the current limitation of the simple two-band model. However, since there is no alternative theory available at this time, we still believe that it is meaningful to start with the two-band model that indeed explains the 3D perovskite successfully. Our experimental work newly demonstrates that the model can fail to explain the novel hybrid perovskites crystallized with 2D layered structures. We do hope to initiate more theoretical approach to this profound problem in the future.

REVIEWERS' COMMENTS:

Reviewer #3 (Remarks to the Author):

The authors have answered my questions.

I suggest the authors add the points in answer 1 to the supporting information for the general knowledge of the readers as why powder measurements would give a more consistent measurements over the thin film.

I recommend the paper to be accepted in Nature Communication after this small modification.

Response to the third referee's report

Reviewer 3's Comments

“The authors have answered my questions. I suggest the authors add the points in answer 1 to the supporting information for the general knowledge of the readers as why powder measurements would give a more consistent measurements over the thin film. I recommend the paper to be accepted in Nature Communication after this small modification.”

Author's Response

As suggested, we have added the points of the reviewer in Supporting Information.